# A Case of In Situ Phage Therapy against *Staphylococcus aureus* in a Bone Allograft Polymicrobial Biofilm Infection: Outcomes and Phage-Antibiotic Interactions

**DOI:** 10.3390/v13101898

**Published:** 2021-09-22

**Authors:** Brieuc Van Nieuwenhuyse, Christine Galant, Bénédicte Brichard, Pierre-Louis Docquier, Sarah Djebara, Jean-Paul Pirnay, Dimitri Van der Linden, Maya Merabishvili, Olga Chatzis

**Affiliations:** 1Pediatric Department, Institute of Experimental and Clinical Research (IREC/PEDI), Université Catholique de Louvain—UCLouvain, B-1200 Brussels, Belgium; 2Department of Pathology, Institute of Experimental and Clinical Research, Cliniques Universitaires Saint-Luc, Université Catholique de Louvain—UCLouvain, B-1200 Brussels, Belgium; christine.galant@uclouvain.be; 3Department of Pediatric Haematology, Department of Pediatrics, Cliniques Universitaires Saint-Luc, Université Catholique de Louvain—UCLouvain, B-1200 Brussels, Belgium; benedicte.brichard@uclouvain.be; 4Department of Orthopedic Surgery, Cliniques Universitaires Saint-Luc, Université Catholique de Louvain—UCLouvain, B-1200 Brussels, Belgium; pierre-louis.docquier@uclouvain.be; 5Center for Infectious Diseases ID4C, Queen Astrid Military Hospital, B-1120 Brussels, Belgium; sarah.djebara@mil.be; 6Laboratory for Molecular and Cellular Technology, Queen Astrid Military Hospital, B-1120 Brussels, Belgium; jean-paul.pirnay@mil.be (J.-P.P.); maia.merabishvili@mil.be (M.M.); 7Pediatric Infectious Diseases, General Pediatrics Department, Cliniques Universitaires Saint-Luc, Université Catholique de Louvain—UCLouvain, B-1200 Brussels, Belgium; dimitri.vanderlinden@uclouvain.be (D.V.d.L.); olga.chatzis@uclouvain.be (O.C.)

**Keywords:** phage therapy, bacteriophage, *Staphylococcus aureus*, chronic osteitis, polymicrobial biofilm, antibiotic-bacteriophage combination, phage-antibiotic interactions, phage-antibiotic synergy

## Abstract

Phage therapy (PT) shows promising potential in managing biofilm infections, which include refractory orthopedic infections. We report the case of a 13-year-old girl who developed chronic polymicrobial biofilm infection of a pelvic bone allograft after Ewing’s sarcoma resection surgery. Chronic infection by *Clostridium hathewayi*, *Proteus mirabilis* and *Finegoldia magna* was worsened by methicillin-susceptible *Staphylococcus aureus* exhibiting an inducible Macrolides-Lincosamides-Streptogramin B resistance phenotype (iMLSB). After failure of conventional conservative treatment, combination of in situ anti-*S. aureus* PT with surgical debridement and intravenous antibiotic therapy led to marked clinical and microbiological improvement, yet failed to prevent a recurrence of infection on the midterm. This eventually led to surgical graft replacement. Multiple factors can explain this midterm failure, among which incomplete coverage of the polymicrobial infection by PT. Indeed, no phage therapy against *C. hathewayi*, *P. mirabilis* or *F. magna* could be administered. Phage-antibiotic interactions were investigated using OmniLog^®^ technology. Our results suggest that phage-antibiotic interactions should not be considered “unconditionally synergistic”, and should be assessed on a case-by-case basis. Specific pharmacodynamics of phages and antibiotics might explain these differences. More than two years after final graft replacement, the patient remains cured of her sarcoma and no further infections occurred.

## 1. Case Presentation

We report the case of a salvage therapy using phages (short for bacteriophage viruses) in a polymicrobial pelvic bone allograft chronic infection (Figure 1). A 10-year-old girl was diagnosed with Ewing’s sarcoma of the left iliac wing in 2015. According to the Euro-Ewing (EE2008) protocol, she received an induction chemotherapy with six cycles of VIDE chemotherapy (vincristine, ifosfamide, doxorubicin and etoposide). A left hemipelvectomy (resection of Enneking’s zones 1, 2 and 4) with femoral head preservation and allograft reconstruction was performed. The anatomopathological examination of the surgically resected tumor showed adequate safety margins and 8–9% tumoral cell viability (persistence of several scattered foci of Ewing’s sarcoma). Post-operative chemotherapy was administered, consisting of 8 cycles of VAI chemotherapy (Vincristine, D-actinomycin, ifosfamide). Given the pelvic location of the lesion and the persistence of viable tumoral tissue, the indication of adjuvant radiotherapy (45 gy) was retained.

Seven months after initial surgery, osteonecrosis of the left femoral head was diagnosed. Eleven months after initial surgery, given the persistence of a dehiscence of the wound, a first surgical revision was performed. Perioperative microbiological samples were positive for *Clostridium hathewayi*. No antibiotic treatment was administered. A recurrence of effusion was observed 4 months after the surgical revision. As there was no growth of microorganism on peripheral swabs, treatment was conservative.

Twenty-four months after initial surgery, the patient presented an acutization of a chronic infection of the allograft, manifested by an increase of the local pus discharge and the detection of *Staphylococcus aureus* in a peripheral swab culture. A second revision surgery with debridement of the allograft was performed, and a vacuum-assisted closure (VAC) system was implemented. Culture of the intraoperative biopsy of the allograft grew *C. hathewayi* and a methicillin-sensitive *S. aureus* (MSSA) with an inducible macrolides-lincosamides-streptogramin B resistance (iMLSB) phenotype. After the surgical debridement, the patient received intravenous flucloxacillin and clindamycin for 5 days. At hospital discharge, antibiotic therapy was prolonged with oral clindamycin and rifampin. This treatment was planned to be continued for 12 weeks. Unfortunately, the rifampin treatment was quickly stopped by the patient after only 5 days due to poor clinical tolerance. Clindamycin was continued alone and stopped after 32 days, because the surgical wound exhibits new inflammatory features with moderate swelling and pus effusion.

Given the therapeutic failures of conventional treatments we decided to consider phage therapy (PT) before considering surgical graft replacement. We obtained the patient’s and her parent’s informed consent with regard to PT in accordance with the conditions of article 37 of the Declaration of Helsinki’s article 37 (“unproven interventions in clinical practice”) [1]. We collaborated with the Queen Astrid Military Hospital (QAMH, Brussels, Belgium), which provided phage preparations and expertise.

Thirty-two months after the first surgery, a third revision surgery was performed, in order to obtain microbiological material for further analysis and for phage susceptibility testing. Pus swabs cultures were positive for *Proteus mirabilis* and *S. aureus.* Pelvic bone biopsies cultures were positive for *C. hathewayi*, *Finegoldia magna*, *P. mirabilis* and *S. aureus.* Among these four bacterial species, the first three are likely organized in a chronic biofilm while the superinfection by the latter, *S. aureus*, triggers acute episodes leading to the inflammatory clinical features newly observed. The patient had not been taking any antimicrobial medication for over six months.

Out of the four identified pathogens responsible for the patient’s infection, the QAMH could only provide us with phages targeting *S. aureus* at that time. Indeed, it only produced phage cocktail BacterioFaag Cocktail 1 (BFC1), which was composed of one anti-*S. aureus* phage (phage ISP) and two anti-*Pseudomonas aeruginosa* phages (phage PNM and phage 14/1) [2]. Phage susceptibility testing (phagogram) was performed on the *S. aureus* strain isolated from pelvic bone biopsies culture using a standard agar overlay plaque assay and showed potent lytic activity of phage ISP against the patient’s *S. aureus* strain, illustrated by an efficiency of plating (EOP) value of 0.7. Unfortunately, no phages effective against *P. mirabilis*, *F. magna* or *C. hathewayi* could be obtained.

Thirty-four months after initial surgery, we initiated this new therapeutic approach combining intravenous antibiotics, surgical debridement and anti-*S. aureus* phage therapy. After surgical debridement of the infected site, the first dose of phages was administered intraoperatively as an in situ instillation of 50 mL of phage cocktail BFC1 at titer 10^7^ Plaque Forming Unit per milliliter (PFU/mL) directly into the surgical site. A “pig tail”-type drainage catheter was installed during procedure for later access to the infected site. In accordance to the last microbiological samples collected, targeted intravenous antibiotic therapy was initiated right after surgical procedure, consisting of clindamycin, rifampin and ciprofloxacin. In situ PT was pursued for 14 days: 40 mL of BFC1 (10^7^ PFU/mL) are instilled through the drainage catheter three times daily during one week, then 30 mL of BFC1 two times daily for an additional week. In order to improve the effectiveness of PT, alkalization of the site was pursued using sodium bicarbonate solution, which was instilled in situ prior to each phage instillation. Bacterial culture of intraoperative biopsies, which were obtained during the debridement before PT initiation, grew *S. aureus*, *P. mirabilis* and *Enterococcus faecalis*. We adapted the antibiotic treatment according to the susceptibilities of the cultured bacteria and replaced clindamycin and ciprofloxacin by intravenous piperacillin-tazobactam.

During PT, the daily re-aspirated sodium bicarbonate samples were sent to the clinical microbiology laboratory for microbiological analysis. While these aspiration cultures showed persistent presence of both *E. faecalis* and *P. mirabilis*, none of them has grew *S. aureus.* In parallel, clinical evolution of the surgical wound site showed signs of improvement over the course of this combined treatment. We noticed a significant decrease in pain, pus effusion and inflammatory signs including a drop in serum C-reactive protein (CRP) levels from 125 mg/L two days after surgical debridement, to 8 mg/L at the end of PT (normal value < 5 mg/L), 14 days after surgical debridement.

This favorable progression led to the patient’s discharge from hospital, twenty days after PT initiation. Outpatient parenteral antibiotic therapy (OPAT) service was organized to pursue antibiotic therapy with oral amoxicillin and rifampin and intravenous ceftriaxone for a planned duration of three months. The patient was followed up as an outpatient. The patient’s clinical condition continued to improve except for the persistence of a serous discharge. Wound swabs were systematically performed and their cultures remained negative for *S. aureus.*

Eighty-two days after PT initiation, the patient declared that she had stopped the oral uptake of both amoxicillin and rifampin for one week, reporting gastro-intestinal adverse events. Full antibiotic treatment had been properly taken for two months instead of the recommended three months.

Four months after PT, resurgence of a significant pus effusion from the wound occurred: for the first time in four months, pus cultures grew *S. aureus.* Recurrence of acute-on-chronic osteitis after both conventional and experimental conservative therapies was an indication to return to *standard of care* management. Accordingly, surgical allograft resection and replacement by a cemented spacer were performed. Empirical antibiotic therapy was started consisting of intravenous ceftazidime, vancomycin and metronidazole. Microscopic examination of intraoperative bone biopsies showed necrotizing osteitis (Figure 2). Bacterial culture results of these intraoperative bone biopsies showed the presence of MSSA/iMLSB *S. aureus* as well as *C. hathewayi* and *Peptoniphilus harei*. Based on these culture results and on a satisfying post-operative evolution, the patient was discharged from hospital and former empirical antibiotic therapy was replaced by OPAT intravenous pipeacillin-tazobactam for a duration of four months.

At the time of writing, more than two years after surgical graft replacement, the now 17-year-old patient is cured of her sarcoma and has known no further infectious episodes. However, she still suffers from orthopedic functional limitations with severe scoliosis.

## 2. Assessment of Phage-Antibiotic Interactions

### 2.1. Introduction

This complex case yielded mixed results after PT was initiated. Marked improvements on both the microbiological and clinical levels were followed by a resurgence of the infection on the midterm. To better understand the potential impact of PT on the patient, we decided to investigate the combined effect of several concentrations of both phage ISP and each of the three anti-*S. aureus* antibiotics that were administered during PT, i.e., clindamycin, rifampin, and ciprofloxacin.

### 2.2. Materials and Methods

To effectively analyze these numerous combinations of effects in an automated, high-throughput and time-dependent way, we used the OmniLog^®^ system (Biolog, Hayward, CA, USA). The use of the OmniLog^®^ system in analyzing phage-bacteria growth kinetics has been previously documented [3,4]. Bacterial growth is monitored in liquid phase at 37 °C, based upon the bacterial cell’s respiration as a universal, standardized marker of bacterial activity. Bacterial cellular respiration reduces a tetrazolium dye, inducing a degree of color change that is measured and registered every 15 min for 72 h in an automated way. Efficacious phages, antibiotics, or combination thereof, should suppress bacterial growth and their measured respiration should decrease accordingly.

We based our analysis on a single isolate of *S. aureus*: the last *S. aureus* isolate retrieved in pelvic bone biopsies before PT initiation, which was the isolate on which the phage susceptibility testing was performed in the QAMH. The *S. aureus* isolate was cultured in lysogeny broth (LB) at 37 °C. Calibration curves were elaborated to reliably correlate optical density at 600 nm wavelength (OD_600_) to colony forming units per milliliter (CFU/mL). The *S. aureus* isolate was grown until reaching OD_600_ of 1.5 on average, which corresponded with 4 × 10^8^ CFU/mL on average. This bacterial titer was validated using a classical plate culture method.

The growth kinetics of the *S. aureus* isolate were assessed in the presence of (i) phage ISP at different values of Multiplicity of Infection (MOI), (ii) each of the three antibiotics that were initiated concomitantly [clindamycin (Fresenius Kabi, Schelle, Belgium), rifampin (Sanofi-Aventis, Gentilly, France), and ciprofloxacin (Bayer, Leverkusen, Germany)] at different concentrations, and (iii) phage-antibiotic combinations. Experiments were done in 96-well plates (MicroWell^®^ Nunclon^®^ Delta, Thermo Fisher Scientific, Rokilde, Denmark) in a final volume of 200 µL of LB supplemented with 100 times diluted tetrazolium dye mix H, according to the manufacturer’s instructions. Bacterial cells were added at a concentration of 10^5^ CFU per well. Ciprofloxacin and rifampin were diluted and tested in the concentration range of 0.00001–10 mg/L, while clindamycin was tested in the range of 0.00001–100 mg/L. Phage ISP was tested at MOI in the range of 0.001–100. The titer of phage ISP was also confirmed after each experiment using the classical double agar overlay method [2]. All combinations were performed in triplicate (biological replicates). All plates contained at least one triplicate of bacterial control wells (containing only 10^5^ CFU of the *S. aureus* isolate in LB broth) and one triplicate of each type of negative control wells (containing only LB broth, only phage ISP and only antibiotics without bacteria).

### 2.3. Results

Omnilog assays showed that phage ISP at MOI ≥ 10.0 completely suppresses the growth of the patient’s isolate for 72 h (see Appendix A). Therefore, results regarding possible synergy could only be observed with lower MOI values (≤1.0) in combination with sub- Minimum Inhibitory Concentration (MIC) concentrations of antibiotics.

Clindamycin suppressed growth of the *S. aureus* isolate for 72 h only at 100 mg/L (see Appendix A). When combined with phage ISP at titer MOI 1.0, sub-MIC concentrations of 10 mg/L of clindamycin induced marked and durable suppression of bacterial growth over 72 h (Figure 3a). Such a level of inhibition could not be achieved by using phage ISP at MOI 1.0 without clindamycin, nor by using any concentration of clindamycin without phage ISP. These results suggest synergistic properties. This synergistic effect was found to be dependent on both high concentration of clindamycin and fixed phage titer of MOI 1: indeed, no similar synergistic effects were observed while using clindamycin concentrations of 1 mg/L or lower, nor while using phage at MOI 0.1.

Rifampin suppressed the growth of the *S. aureus* isolate for 72 h at concentrations ≥0.1 mg/L (see Appendix A), and did not show synergistic effect on *S. aureus* when combined with phage ISP at sub-inhibitory MOIs (Figure 3b). Interestingly, rifampin and ISP combinations even appeared to be slightly antagonistic in specific conditions. Combinations of concentrations of rifampin (0.01 mg/L) and phages (MOI 0.1) appeared less effective at inhibiting bacterial growth than the same concentration of rifampin used alone. This possible moderate antagonism was shown to be concentration-dependent. Sub-inhibitory titers of phage ISP were never found to mitigate the effect of high above-breakpoint concentrations of rifampin, and sub-inhibitory concentrations of rifampin were never found to mitigate the effect of highly inhibitory titers of phage ISP in our OmniLog^®^ assays.

Ciprofloxacin suppressed the growth of the *S. aureus* isolate for 72 h at concentrations ≥1.0 mg/L (see Appendix A). At lower concentrations, when combined to phage ISP at two different MOIs of 1.0 and 0.1, a slightly increased inhibition of growth was observed in comparison to the use of ciprofloxacin or phage ISP only, at the same concentrations (Figure 3c). However, as opposed to the more marked synergistic properties of clindamycin, ciprofloxacin and phage ISP combination could never fully suppress bacterial growth, and eventually reached similar levels of detected bacterial activity as the bacterial control after 72 h (see Appendix A).

## 3. Discussion

This complex case of a bone allograft polymicrobial osteomyelitis highlights the difficulties of treating this type of infection with conventional therapy, but also with PT. Multiple causes could explain the therapeutic failure of conventional treatment: (i) the chronicity of the *C. hathewayi* infection, with an infection that has been detected for more than two years at the time of PT initiation, (ii) the acutization of the infection with *S. aureus*, a bacterial species known to create biofilms and therefore difficult to treat with conventional antibiotics, (iii) the polymicrobial nature of the osteitis requiring a combination of several antibiotics, leading to poor clinical tolerance, and (iv) the acquisition of antibiotic resistance mechanisms. In addition, local postoperative radiation therapy could have hindered antibiotic pharmacokinetics by limiting their distribution to the infected site. Indeed, radiation-induced fibrosis has been reported to deteriorate microvascular circulation in both bone and skin and soft tissues, which could result in poorer in situ distribution and effect of intravenous antibiotics [5,6].

While all cultured *S. aureus* isolates showed a MSSA phenotype, they were also exhibiting an iMLSB phenotype, which was assessed by D-test [7]. Detection of this iMLSB phenotype was considered a relative contraindication to the reintroduction of clindamycin, an antibiotic of the lincosamides family. Indeed, its administration could induce rapid resistance to MLSB antibiotics in *S. aureus*, which could result in therapeutic failure and loss of valuable therapeutic options. These reasons made us ultimately consider PT as last resort conservative treatment.

Introduction of PT in addition to antibiotics and surgical debridement yielded contrasting results in this patient. On the one hand, the situation made promising progress on the short term after PT initiation, on both the microbiological and the clinical plan. On the microbiological plan, culture of deep liquid aspirations samples from the infected wound remained negative for *S. aureus*, which the PT specifically targeted, during the whole course of PT and up to at least two months after it was discontinued. On the clinical plan, PT initiation correlated with a rapid decrease in local and biological inflammatory features with favorable evolution of CRP levels from 125 mg/L after 48 h of PT to 8 mg/L after 14 days of PT (normal values < 5 mg/L).

However, mid-term progression was less favorable. Three months after PT discontinuation, recurrence of the infection was diagnosed. Pus culture were positive for a similar phenotype of *S. aureus.* This could suggest a relapse of the infection caused by the same strain as the one we isolated before PT initiation, yet does not totally rule out the possibility of a reinfection with a different *S. aureus* strain that would harbor the same MSSA/iMLSB phenotype.

Several factors could explain the post-PT relapse of infection in our patient. Compliance with post-operative antibiotic therapy was poor, antibiotic intake being unilaterally discontinued by the patient after two months instead of the recommended three months. Moreover, the postulate under which PT was initiated was likely flawed. Although the infection in our patient was polymicrobial, the QAMH could only provide phages targeting the patient’s *S. aureus* strain. Despite an extensive search, contacting foreign phage research laboratories and institutes, we were unable to obtain phages against *C. hathewayi*, *F. magna*, or *P. mirabilis*. Nevertheless, we initiated PT against *S. aureus* only, believing that eradicating the more acute and aggressive pathogen could have an impact on the broader organization of the chronic polymicrobial biofilm. This “kill-the-leader” strategy seemed successful at first, but did not pay off on the long term. We hypothesize that part of the polymicrobial infection and its biofilm configuration were maintained, and eventually remained a suitable terrain for new *S. aureus* superinfection. This eventually brought the situation back to its initial dead-end after four months of respite. Recent attempts at treating orthopedic infections using PT suggest that this therapy should be used in combination with antibiotics and proper surgical care; they also seem to achieve durable clinical and microbiological successes by focusing on mono- or dual-species biofilms where full phage coverage of the pathogens can be achieved [8,9,10,11]. In our case, full coverage of the polymicrobial infection by PT was not possible.

The analysis of this case suffers from a number of limitations. First, we did not keep any *S. aureus* isolate retrieved after PT, either when it was first detected in pus swab culture upon infection relapse, or when it was detected in the intraoperative samples collected during surgical graft replacement. Comparing phage susceptibility between those isolates and the initial *S. aureus* isolate on which our investigations were based could have highlighted mechanisms of phage-resistance acquisition. We also did not keep any serum sample that would have allowed for the detection of potential phage-directed humoral immune response. However, limiting PT duration to two weeks likely mitigates the risk of immune neutralization development, even though in situ local PT is notably able to induce it past this term [12]. Nevertheless, these reactions should not automatically be considered a hurdle for in situ PT’s efficacy [12].

Assessment of potential interactions between phages and concomitant antibiotics through OmniLog^®^ assays provided interesting results: we will further discuss the case of phage-clindamycin and phage-rifampin interactions.

Adjunction of phage ISP at MOI 1.0 had seemingly synergistic properties with clindamycin, possibly conferring in vitro re-sensitization to high clindamycin concentrations of 10 mg/L. A potential explanation to this phenomenon might be found at the molecular and metabolic level in *S. aureus.* As explained, this *S. aureus* strain exhibits resistance to clindamycin through an iMLSB phenotype. This phenotype is caused by the expression of mostly plasmid-borne *erm*-family genes, encoding for Erm-family methyltransferases that confer protective methylation to the molecular target of macrolides, lincosamides and streptogramin B: the 23S ribosomal RNA (rRNA) of the larger 50S ribosomal subunit. In the iMLSB phenotype, inducibility mechanisms towards the *erm-*family genes reside at the post-transcriptional level. Schematically, the *erm-*family gene is transcribed in a cognate mRNA molecule whose ribosomal translation is then blocked by a “leader peptide”, preventing the production of methyltransferases; “inducer” molecules, such as erythromycin, can in turn block the leader peptide’s translation, which allows unbridled translation of the *erm-*mRNA into methyltransferases, leading to the development of iMLSB resistance phenotype [13]. Though it has been subject to contradicting claims, recent experiments show that erythromycin is not the only antibiotic able to induce Erm methyltransferase production through leader peptide repression: clindamycin itself, *inter alia*, can exert similar activity on the ErmB methyltransferase-ErmBL leader peptide couple, though not in the exact same way as erythromycin [13,14,15]. Consistent with this hypothesis, no concentration of clindamycin alone could suppress peak bacterial growth in OmniLog^®^ culture, suggesting clindamycin-resistance had been induced by clindamycin itself. In the patient’s *S. aureus* strain, clindamycin possibly induces unbridled translation of *ermB-*mRNA into ErmB methyltransferases, rapidly leading to the acquisition of a clindamycin-resistant phenotype. Yet, adjunction of phage ISP at MOI 1.0 seems to reduce the resistance-inducing power of clindamycin, leading to a marked and durable suppression of bacterial growth. This could be explained by the fact that phage ISP, as a virus, diverts ribosomal activity for its own needs of progeny virion production and completion of lytic cycle [16]. This subversion of the ribosomal function can have dramatic effects on bacterial protein translation, at times suppressing it entirely [17]. In our case, phage ISP-mediated subversion of bacterial ribosomal function possibly prevented translation of *ermB-*mRNA, even though clindamycin had likely released it from ErmBL sequestration. This would result in reduced or suppressed ErmB methyltransferase activity, which could explain how phage ISP reduces the resistance-inducing properties of clindamycin in certain iMLSB *S. aureus* strains.

Conversely, combination of phage ISP with rifampin did not result in synergistic effect. Furthermore, breakpoint concentrations of rifampin combined with sub-inhibitory phage titer of MOI 0.1 resulted in a moderately weaker growth-inhibitory effect than the same breakpoint concentration of rifampin used alone. Molecular basis for antagonistic pharmacodynamics between phages and rifampin do exist. Rifampin’s bactericidal properties are based on its inhibitory properties towards bacterial RNA polymerase [18]. However, phage ISP does not encode for its own RNA polymerase, and fully depends on its bacterial host’s RNA polymerase for transcription [19]. Rifampin’s effect on ISP’s bacterial host could thus prevent phage genome transcription, resulting in antagonistic effects. Consistent with this hypothesis, rifampin’s inhibitory effect on some phages’ RNA synthesis had already been documented decades ago [20]. More recently, a literature review presented numerous occurrences of pre-clinical studies documenting rifampin’s inhibitory effects on phage infectivity [21]. Yet, evidence suggests that the possibility of phage-rifampin synergy should not be ruled out in specific conditions, for example through combined use of rifampin and phage Sb-1, another representative of genus *Kayvirus*, in rifampin-susceptible *S. aureus* biofilms [22].

Finally, we should acknowledge that although these two phage-antibiotic interactions have likely taken place in vitro, whether they played any role in vivo regarding the case’s progression remains uncertain. Several limitations of interpretation shall be noted. First, OmniLog^®^ assays analyzed the growth of a single bacterial species, *S. aureus*, in short-term (72 h) broth culture. To what extent the results obtained from these assays can be transposed to the pharmacodynamics and pharmacokinetics that took place in a chronic, biofilm-organized polymicrobial infection is unknown. Similarly, interactive properties in our OmniLog^®^ assays depend on both specific antibiotic concentrations and phage titers, the combination of which might not have been achieved in vivo. As all reagents were added to the microplates simultaneously, our OmniLog^®^ assays did not investigate the effect of different chronological sequences of application of phage ISP and antibiotics to the bacterial culture. This limitation warrants further investigation, as recent in vitro evidence advocating phage-antibiotic synergies in *S. aureus* biofilm strongly suggests that different sequences of application of phages and antibiotics can result in significantly different bactericidal effects, favoring a “phage first” approach for staggered phage-antibiotic administrations [23]. Recent clinical application of these findings came to the same conclusion [11].

## 4. Conclusions

Combination of in situ anti-*S. aureus* phage therapy, intravenous antibiotics and surgical debridement was used as a salvage therapy for a polymicrobial chronic osteitis of a pelvic bone allograft. This led to favorable short-term results with regard to both the microbiological and clinical level, yet failed to prevent a recurrence of infection on the midterm, which led to surgical graft replacement. The reasons for this midterm failure are multifactorial, including poor compliance with antibiotics and incomplete coverage of the polymicrobial infection by phage therapy. Phage-antibiotic interactive properties should not be considered “unconditionally synergistic”, and should be assessed on a case-by-case basis, taking into account antibiotic type, antibiotic concentration and phage titer. Specific pharmacodynamics of phages and antibiotics might explain these differences. Further research is needed to clarify the relevance of these findings in clinical practice.

## Figures and Tables

**Figure 1 viruses-13-01898-f001:**
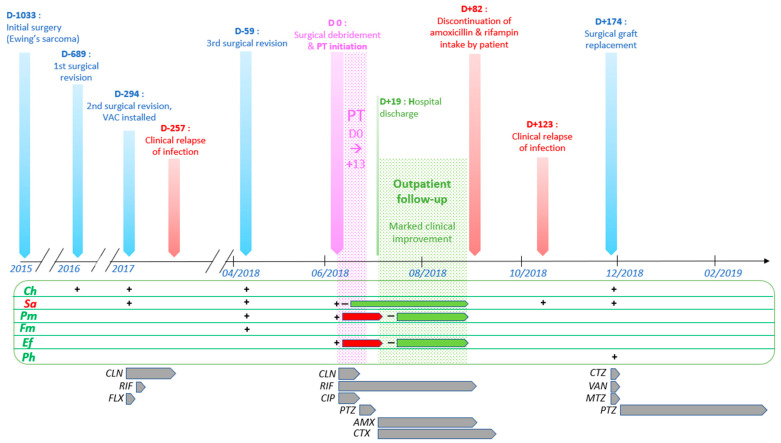
Timeline of the most relevant events, procedures and treatments in the patient’s history, from initial Ewing sarcoma surgical resection to termination of antibiotic therapy after final surgical graft replacement. From top to bottom: vertical arrows represent important events in the clinical progression of the case. Blue vertical arrows represent surgical procedures. *VAC: vacuum-assisted closure; PT: phage therapy.* Microbiological culture results are displayed in the green horizontal box. Horizontal arrows in this box represent continuations of similar results (red: positive; green: negative) over repeated daily analysis. *Ch: Clostridium hathewayi; Sa: Staphylococcus aureus; Pm: Proteus mirabilis; Fm: Finegoldia magna; Ef: Enterococcus faecalis; Ph: Peptoniphilus harei.* Lower horizontal arrows represent antibiotic therapies. *CLN:* clindamycin; *RIF:* rifampin; *FLX:* flucloxacillin; *CIP:* ciprofloxacin; *PTZ:* piperacillin-tazobactam; *AMX:* amoxicillin; *CTX:* ceftriaxone; *CTZ:* ceftazidime; *VAN:* vancomycin; *MTZ:* metronidazole.

**Figure 2 viruses-13-01898-f002:**
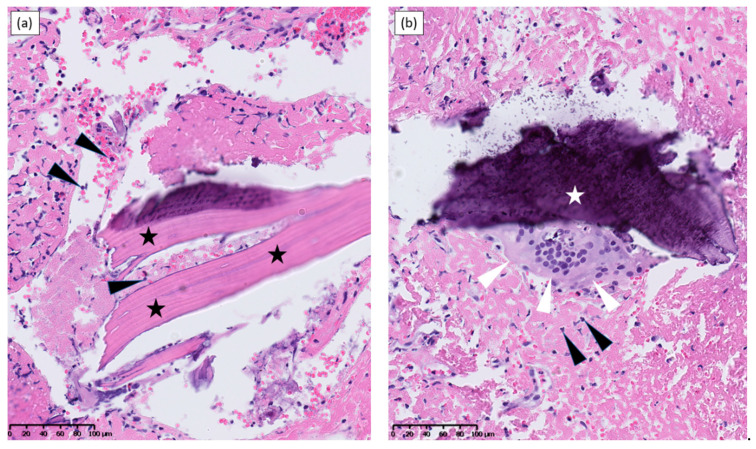
Pathological findings from pelvic bone allograft biopsies obtained upon infection recurrence, six months after phage therapy has been discontinued. (**a**) Lamellar bone fragments (stars) appear necrotic, displaying only empty osteocyte lacunae; immediate surroundings are infiltrated with inflammation-inducing figurative elements of blood with numerous leucocytes (arrows). (**b**) Necrotic bone fragment (star) is lysed by a massive osteoclast (white arrows). Immediate surroundings are again infiltrated with numerous leucocytes (black arrows).

**Figure 3 viruses-13-01898-f003:**
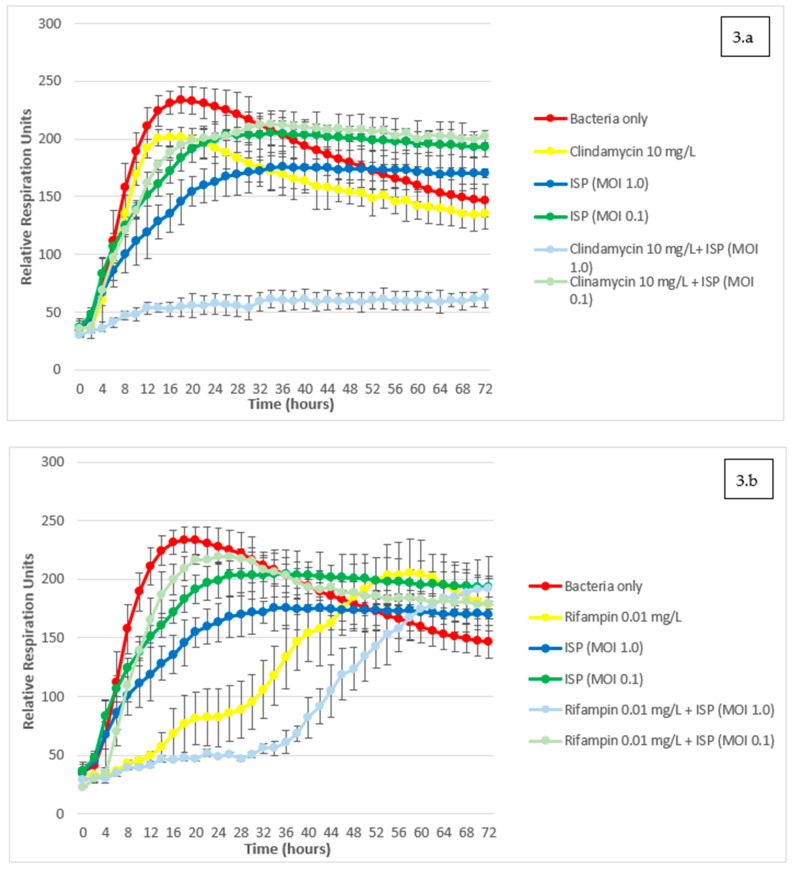
Assessment of phage-antibiotic interactions with the OmniLog^®^ system. Kinetics of bacterial growth under various conditions are expressed as Relative Respiration Units over a 72 h timeframe. Phage ISP at various Multiplicities of Infection (MOI) was combined with different sub-MIC (Minimum Inhibitory Concentration) concentrations of the three antibiotics that were used concomitantly in the patient: (**a**) clindamycin; (**b**) rifampin; (**c**) ciprofloxacin. Results are presented as mean values of three experiments (biological replicates) with error bars representing standard deviations of the means.

## Data Availability

Not applicable.

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
