# Peer review of "A Case of In Situ Phage Therapy against Staphylococcus aureus in a Bone Allograft Polymicrobial Biofilm Infection: Outcomes and Phage-Antibiotic Interactions"

_viruses, 2021, doi:10.3390/v13101898_

Round 1
Reviewer 1 Report
English Language and style are minor spell check required
Author Response
We thank the editor and the reviewers for their precious time in reviewing our paper and providing their recommendations and comments.
English spelling and style have been double-checked with several co-authors and modified in the new version of the manuscript where deemed necessary. Thank you for this suggestion.
We hope that the current revised version of the manuscript will be to your entire satisfaction.
Best regards,
Brieuc Van Nieuwenhuyse
Reviewer 2 Report
Van Nieuwenhuyse and colleagues describe a case of phage therapy applied to a 10 year old patient with a multispecies chronic infection in a bone allograft, which resulted in mixed effects at short and medium term. The authors further assessed phage-antibiotic interactions between the phage and antibiotics administered during PT, and report distinct effects depending on the antibiotic used. While some of the observations have been reported previously, this study is valuable and provides important information for the use, do’s and dont’s of phage therapy. There are a few issues that I believe the authors should address before publishing, as detailed below.
Major comments
- Title could be simplified
- Abstract, lines 36-37. The sentence as it is seems to imply that PT was the responsible for the patient’s cure, which is not the case. Please remove or rewrite sentence.
- Could the authors provide more details about the rationale for phage therapy dosage, schedule of administration, and use of in situ application? (lines 110-112, 116-117). Also, why was phage therapy administered only for 2 weeks?
- Was targeted intravenous antibiotic therapy experimented prior to phage therapy? If not, how do the authors know that improvements of the patient’s condition was not solely related to this targeted antibiotic therapy rather than the phages? If this cannot be ascertained, then I would advise that the authors mention both PT and targeted antibiotic therapy when discussing the signs of improvement observed.
- How have the authors determined that the effect observed from the combination of phage ISAP and clindamycin is synergistic and not additive?
- When assessing phage-antibiotic interactions, the authors tested different phage and antibiotic concentrations, but how do these related to those used in the patient? Especially for the phage, the authors assess specific MOIs, which are determined as the ratio between phage and bacteria concentrations. In the patient I believe the concentration of bacteria was unknown, and therefore the MOI of the phage was also unknown.
- For Figure 3, the negative control (control – broth only) should be subtracted to the remaining samples, and therefore not plotted. I also suggest that the authors improve the visual of the graphics, for example by adding a vertical line in the y axis, remove the internal horizontal lines, maintain consistency between graphics. XX axis title should be ‘Time (hours)’. Please align the graphs (e.g. (b) is above (a)). Why is MOI of 1 used for (a) and an MOI of 0.1 used for (b) and (c)? If the authors chose to represent these MOIs in the graphics because they provided the best results, they should mention this in the legend and provide all additional results as supplementary data. Furthermore, I note that the effect of phage ISP at MOI of 0.1 in graphics (b) and (c) is distinct, although they should provide similar results. In fact, the curve of ISP in (c) more closely resembles the curve in (a) using MOI of 1.0. It is also interesting to note the very poor effect the phage has on bacterial growth even at an MOI of 1.0, and it does not seem to be an efficient phage. The authors should depict standard deviations of the curves, to better assess for assay variations and significance of the results.
Minor comments/questions
- Lines 98-99, why was the patient not taking any antibiotics for 6 months if the infection was present?
- Line 197, I believe the authors mean phage ISP and not phage M1.
Reviewer 3 Report
Authors provide a case report of phage-antibiotic-surgical combined therapy. in my opinion, phage-antibiotic synergy should be calculated and presented numerically this will provide more information. unfortunately, due to date, there is no method that will allow calculating this. maybe this could work:
https://www.liebertpub.com/doi/pdf/10.1089/phage.2019.0001#:~:text=Materials%20and%20Methods%3A%20This%20study,(MOIs)%2C%20and%20MV50%E2%80%94
https://www.sciencedirect.com/science/article/pii/S1369703X20302060
this is a future suggestion.
authors correctly guide the reader through valuable text
Author Response
Dear reviewer,
We would like to thank you for your precious time in reviewing our paper and providing valuable comments and recommendations.
We do agree that a quantification of the synergy's magnitude would be a significant asset to our work. Since another reviewer also recommended that we put some interest into this subject, and even though we are facing a lack of "gold standard" method to reliably quantify phage-antibiotic synergy, we illustrated the synergy's magnitude by performing Bliss independence test at a specific timepoint of +12h for the Clindamycin+ISP assay (which is the only one where we claim that the combined effect is indeed synergistic).
This test determines the expected additive effect of a drug combination based on a non-interaction hypothesis (additive combination, absence of synergy). This expected additive combined effect (Rc) is determined by:
Rc = R1 + (1−R1) × R2
Where R1 is the response to drug 1 (phage ISP at MOI 1,0) and R2 is the response to drug 2 (clindamycin 10 mg/L). At timepoint +12 hours, and after subtracting negative control (bacterium-free wells) values to all other values, the drug responses expressed in relative reduction in bacterial respiration units (compared to bacterial control values) are:
R112h = 0.507
R212h = 0.071
Applying the abovementioned Bliss independence test formula gives us the expected additive effect of this drug combination under the assumption of no interactions:
Rc12h = 0.542
The Combination Index (CI) can be obtained by dividing this expected result by the real combined effect that we observed (Ro12h):
Ro12h = 0.933
CI12h = Rc12h / Ro12h = 0.581
Synergy being confirmed by a CI inferior to 1.
Note: we did not include these calculations in the new version of the manuscript ; please tell us if you would like us to include them.
Regarding the other two antibiotics, statements including the words "synergy" or "synergistic" will be modified to avoid confusion.
References :
- Roell, K.R.; Reif, D.M.; Motsinger-Reif, A.A. An Introduction to Terminology and Methodology of Chemical Synergy—Perspectives from Across Disciplines. Frontiers in Pharmacology 2017, 8, doi:10.3389/fphar.2017.00158.
- Ma, J.; Motsinger-Reif, A. Current Methods for Quantifying Drug Synergism. Proteom Bioinform 2019, 1, 43-48.
Though admittedly imperfect, we hope this complementary explanation will be to your satisfaction. As you suggested, we will definitely keep in mind your methodological recommendations so as to try to evaluate such synergies in a more standardized and robust way in the future.
Thank you again for your time and consideration.
Best regards,
Brieuc Van Nieuwenhuyse